# Have Sanctions Modified Iran's Trade Policy? An Evidence of Asianization and De-Europeanization through the Gravity Model

**Liudmila Popova and Ehsan Rasoulinezhad \***

Department of World Economy, Faculty of Economics, Saint Petersburg State University,
St. Petersburg 191123, Russia; l.v.popova@spbu.ru
\* Correspondence: erasolinejad@gmail.com; Tel.: +7-921-866-0980

**Abstract:** This study is an empirical attempt to find out whether under sanctions Iran's trade direction has shifted away from Europe (trade policy of de-Europeanization) towards Asia (trade policy of Asianization). The analysis is conducted using a panel-gravity trade model to analyze bilateral trade pattern between Iran and 50 countries from the EU and Asia during the period 2006–2013. To this end, the authors use an extended gravity model by adding new variables, including the index of Chinn–Ito (KAOPEN) as an indicator of financial openness, and the composite trade intensity (CTI) as an indicator of trade openness. Our findings reveal that the gravity equation fits the data reasonably well. The empirical evidence indicates a significant negative effect of sanctions on Iran–EU bilateral trade (by an average of 46.9%), while it has a positive impact on trade between Iran and the Asian countries (by an average of 85.2%). Overall, these findings confirm that the imposition of various sanctions related to Iran's nuclear program has pushed the country's foreign trade to reorient away from Europe towards Asia.

**Keywords:** sanctions; trade policy; gravity model; Iran

## 1. Introduction

Economic sanctions as a penalty levied on a country are one of the most debated topics in international trade. Although the review of economic restrictions has been the subject of a number of studies in recent years (Askari et al. (2003) [1]), the effect of sanctions on Iran's foreign trade has not drawn sufficient attention of the scholars.

Over the years, Iran as an Asian country, has been placed under increasingly harsh rounds of sanctions imposed by different nations for various reasons. The history of imposing sanctions on this country dates back to the Iran's oil nationalization in 1951 when the Iranian government nationalized the oil industry, kicking out the Anglo-Persian oil Company and setting up the National Iranian Oil Company. Following the chaotic aftermath of Iran's Islamic revolution when militants in Iran seized some American citizens at the embassy of the United States in Tehran (the hostage crisis (1979–1981)), the United States imposed a new round of sanctions against Iran. The last round of sanctions started in response to the Iranian nuclear program in 2006 included harsh restrictions such as disconnection of Iranian banks from the SWIFT (Society for Worldwide Interbank Financial Telecommunication) system and the EU oil embargo. Since these tough restrictions have had an adverse impact on isolating and hindering Iran's economic development, the round is considered as the worst one against Iran.

In order to minimize adverse consequences of sanctions, Iran began to implement a policy of import substitution and modified her foreign trade strategy. The implementation of import substitution

policy passed by the Iranian Parliament's Committee on Economy in the early of 2000s has enabled the country to reduce her dependence on imports of some goods (e.g., medicines), balance her local supply and improve various industries in order to produce non-oil exportable commodities. Deep reduction in trading volume with Europe due to the sanctions has prompted Iran to reorient her foreign trade away from Europe to new markets in order to prevent an economic collapse and ensure the viability of the import substitution policy. Since this reorientation has led primarily to significant expansion of Iran's trade volume with Asian nations, particularly with China and India, as well as promotion of trade ties with other countries in greater Asia, it was named the policy of Asianization and de-Europeanization of Iran's foreign trade.

To the best of our knowledge, this paper is the first to study the Iran's trade policy of Asianization and de-Europeanization during the period of sanctions. Although the changes of foreign trade flow of Iran have drawn some attention from researchers such as Rasoulinezhad (2016) [2], Fahimifard (2013) [3], Suvankulov and Guc (2012) [4], Soori and Tashkini (2012) [5], Taghavi and Hosein Tash (2011) [6], Esmaeli and Pourebrahim (2011) [7] and Kalbasi (2002) [8], we did not find any study that addressed the consequences of sanctions on Iran's bilateral trade with the EU members and Asian countries through a gravity model. To this end, a gravity model is applied using data of the bilateral trade between Iran-25 EU members and -25 Asian countries (the choice of these 50 countries in the study is based on the full list of all Iran's trading partners and their ranking by IRICA[1]) for the period from 2006 to 2013. Thus, the comparison of the trade patterns between Iran and these nations in Asia and the EU may help to draw a conclusion about whether economic sanctions against Iran have forced the country to modify her trade policy and move forward to Asianization and de-Europeanization.

Following the objective of the research, assumptions on the gravitational theory and consideration of the sanctions as an influential factor in modifying Iran's trade pattern, the research null hypotheses are:

(i)　　There is a negative relationship between the sanctions against Iran and Iran–EU members trade.
(ii)　　There is a positive relationship between the sanctions against Iran and Iran–Asian countries trade.

The remainder of this research is structured as follows. The next section provides a theoretical framework. After that, a brief literature review is presented. Data and methodology are discussed in the fourth section. Then, research results are presented and the last section concludes with a discussion and directions for further research.

## 2. Theoretical Framework

Caruso (2005) [9] indicated that economic sanctions may act as quantitative restrictions. Based on his argument, Figure 1 represents the impact of sanctions on bilateral trade volume and prices for two countries—the sender and the target. By considering a target country's import demand curve, D, and assuming a target country to be a small open economy (supply curve is a flat line), the pre-sanctions equilibrium will be at point E. The imposition of trade sanctions by a sender country to the target forces the export volume to move to $q^*$ and import prices raise to $p^*$ as well. The wedge in the price $(p^* - p_w)$ indicates the quantitative restriction and a rent that can be distributed to the government or to the private agents equals $(p^* - p_w) \times q^*$.

Following the argument of Kaempfer and Lowenberg (1992 [10], 1999 [11]), we can explain the consequences of trade sanctions as well. Suppose in the global trade market a country T (a potential target of economic sanctions) trades with her trade partners in the world, W. Implementation of multilateral economic sanctions by W pushes the target to move from her trade equilibrium to the autarky mode. This situation rarely happens in the real world because unilateral economic sanctions are more likely to happen to a target. By imposing the economic unilateral sanctions on a target by

---

[1]　　The Islamic Republic of Iran Customs Administration.

a sender, the trade volume of the target reduces in the short-run because the number of her trade partners (W–S) becomes smaller. If other trade partners join the sender to support the sanctions regime (meaning a larger number of senders relative to non-sanctioners), the magnitude of reduction in trade volume may be greater. As the number of senders increases, trade goes to the autarky. However, if we assume that there are alternative markets for T, the target can shift her trade destinations from a sender (s) to the new ones. Hence, we can conclude both the sender and the target hit by trade sanctions. Moreover, in the global trade market, non-senders can obtain additional gains by improving trade relations with the target country. It should be mentioned that, as Van Bergeik [12] argues, transportation cost is an important factor for a target country that wants to find new trading partners under sanctions. If local exporters incur higher transportation costs to export goods to more distant locations, they would give up trading. Hence, under the heavy sanctions regime, only large private exporters or state-sponsored companies who can pay higher transportation costs, can engage in trade with new partners. On the other hand, Yang et al. (2009) [13] believe that if a target can easily and smoothly change her trade towards new partners, the sanctions of a sender may be neutral or less effective.

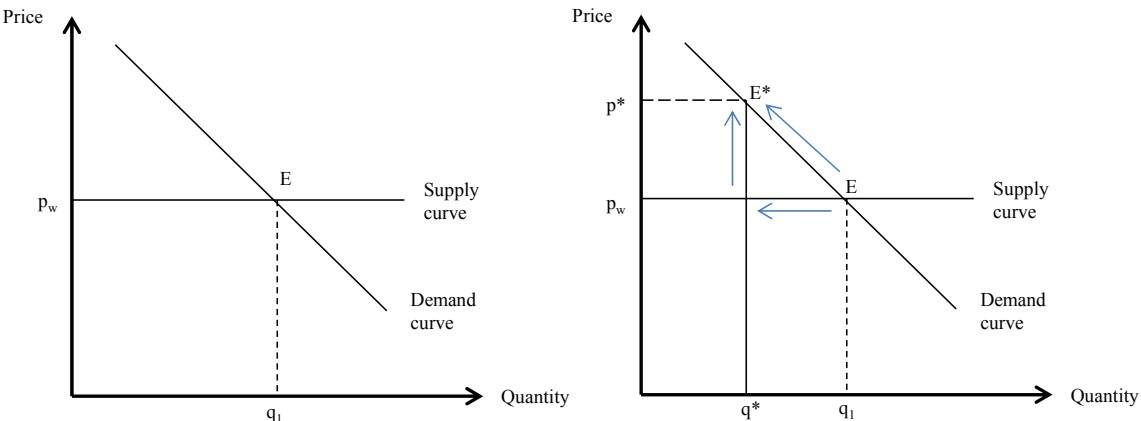

**Figure 1.** The impact of sanctions on trade volume. Source: Compiled by authors from Caruso (2005) [9].

## 3. Literature Review

The related literature can be divided into three strands of study: (i) investigation of sanctions' effects on economic variables of a certain country; (ii) exploring the effects of sanctions by using the gravity model; and (iii) consideration of sanctions' effects on the economy of Iran.

In the first strand of the study, the authors concentrated on various economic and political influences of sanctions. Dollery (1993) [14] tried to make a conceptual note on financial and trade sanctions against South Africa by using a conventional general equilibrium analysis. Under comparative static findings, he showed that for small developing economies, trade sanctions have a greater share of the sectoral burden on the labor-intensive exportables sector. In other study, Evenett (2002) [15] investigated the effect of sanctions on South Africa during the apartheid regime. His findings depicted that exports of South Africa to the EU have recovered rather faster compared to that to the United States. Jafarey and Lahiri (2002) [16] attempted to find out the interaction between credit markets, trade sanctions and the incidence of child labour. The study found evidence of access reduction to credit under sanctions and an increase of the number of child workers, especially among poor households. In other study, Lamotte (2012) [17] explored the effects of wars and sanctions on trade flows of former Yugoslavia during the 1990s. He identified three episodes of war in the former Yugoslavia, two of them associated with the proclamation of independence (by Croatia and Slovenia in 1991, Bosnia and Herzegovina in 1992–1995), and the third war occurred due to the surge of violence in Kosovo in 1998. All of these wars caused the United Nations and the EU to impose sanctions against the Federal Republic of Yugoslavia (FRY). He confirmed that the impact of sanctions on trade

volume is more pronounced than the impact of war. Furthermore, the consequences of both war and sanctions persisted for several years after they ended. Neuenkirch and Neumeier (2015) [18] empirically assessed how economic sanctions imposed by the United Nations and the United States affect the GDP growth in 160 countries over the period 1976–2012. They found that the United Nations sanctions have a significant influence on the target state's economic growth, while the effect of the United States sanctions is much smaller and less distinct. In addition to these studies, a number of scholars drew attention to the impact of sanctions on several indicators such as drug availability in an economy (Karimi and Haghpanah (2015) [19]), poverty (Neuenkirch and Neumeier (2016) [20]), exchange rate (Dreger and et al. (2016) [21]) and income inequality (Afesorgbor and Mahadevan (2016) [22]).

The similar result of most of the studies in the first strand is that sanctions play a significant and influential role in changing economic variables.

The second strand of literature attempted to find out the effect of sanctions using the gravity model. Hafbauer et al. (1997) [23] are some of the first researchers who investigated sanctions' effects through a gravity model. To this end, they ran a gravity trade model based on the global trading patterns for 88 countries in three different years of 1985, 1990 and 1995. Their econometric estimations proved that sanctions reduce bilateral trade flows by nearly 90%. In other study, Yang et al. (2004) [24] used a gravity model to explore the results of economic sanctions. Their findings showed that success of sanctions is most easily achieved when pre-sanction relations between sender and target are cordial or neutral. Caruso (2005) [9] applied a gravity trade model in the case of the United States and 49 target countries over the period 1960–2000. For simplicity, he defined two types of sanctions, i.e., partial trade restrictions and financial sanctions as "limited and moderate", and extensive trade and financial restrictions as "extensive". The findings depicted that extensive sanctions have a larger negative impact on bilateral trade than limited and moderate restrictions. In other paper, Yang et al. (2009) [13] developed the gravity model used by Yang et al. (2004) [24] to find out whether the EU is an alternative market for nations when they are confronted with the United States sanctions. Their main finding expressed that after the imposition of sanctions by the United States, the EU gradually captured trade flows from the targeted countries. Lastly, Mehchy et al. (2015) [25] investigated the effects of sanctions by using a gravity model on Syrian exports between 1995 and 2010. The major finding in their study revealed that sanctions and the deterioration in institutional factors have reduced Syria's export potential by more than 70%.

Overall, the important points of the studies in the second strand are (i) the gravity model is a proper approach to find out the effects of sanctions on trade flows; and (ii) sanctions influence bilateral trade flows between countries.

The third strand of research considered the effects of sanctions on the Iranian economy. However, it should be noted that relatively few studies investigated this subject. Aghazadeh (2014) [26] tried to explore the impact of western multilateral sanctions on the Iran's economy. The main result of his study confirmed a significant impact of sanctions on Iran's macroeconomic indicators, particularly on trade flows. Bazoobandi (2015) [27] investigated the consequences of sanctions on Sino-Iranian relations. The results depicted that the Western sanctions were the driving force behind the development of various aspects of the relations between Iran and China. In other study, Borszik (2015) [28] tried to explain the effects of international sanctions against Iran and domestic responses of the country on the power structure of the targeted regime. The research results showed that international sanctions were initially harmful to the Iranian economy, but as long as the country used some strategies such as unofficial financial networks and finding new economic partners, her economic system has gradually become immune with the countering the economic sanctions. Haidar (2016) [29] investigated the relationship between sanctions and export deflection in Iran over the period 2006–2011. The main results concluded that two-thirds of the Iranian export volume deflected towards the non-sanctioning countries.

Overall, it seems that there was no serious attempt to examine whether under the pressure of sanctions Iran has modified her trade pattern. Hence, this paper will provide new and useful insights about how various factors, such as sanctions, can affect the trade pattern of Iran with different nations, especially with the EU member states and Asian countries.

## 4. Data and Methodology

### 4.1. Dataset Description

This study covers bilateral trade between Iran and her trade partners which consists of 25 EU member states (Austria, Belgium, Cyprus, Czech Republic, Denmark, Estonia, Finland, France, Germany, Greece, Ireland, Italy, Hungry, Latvia, Lithuania, Luxembourg, Malta, The Netherlands, Poland, Portugal, Slovakia, Slovenia, Spain, Sweden and the United Kingdom[2]) and 25 Asian countries (Afghanistan, Bangladesh, China, India, Indonesia, Iraq, Japan, Kuwait, Lebanon, Malaysia, Nepal, Oman, Pakistan, Philippines, Singapore, South Korea, Syria, Taiwan, Tajikistan, Thailand, Turkey, Turkmenistan, UAE, Uzbekistan and Vietnam) over the period 2006 to 2013. The variables used in this study which are shown in Table 1, contain aggregate trade volume[3] (sum of import and export) between Iran and these countries in thousands of U.S. dollars, GDP and GDP per capita in thousands of U.S. dollars, distance between Iran and the trade partners in kilometers, the Chinn–Ito index (KAOPEN) and the composite trade intensity (CTI) in percent, GDP weighted average of distance as a proxy for Multilateral Resistance Term (MTR) and sanctions as a dummy variable. The source of the data on aggregate trade volume is IRICA [31]. The data on GDP, GDP per capita and primary data to calculating CTI are collected from the World Bank [32] and the IMF [33]. Data for distance between countries were gathered from CEPII [34] and the Chinn–Ito index data were collected from the Portland State University website[4].

Furthermore, all the time-variant series level are transformed in to natural logarithms, based on the advantages of this form than using the level of variables (Wooldridge (2013) [35]).

**Table 1.** The variables of model.

| Variables | Definition | Unit |
|:---:|:---:|:---:|
| *Trade* | Aggregated trade volume between Iran and trade partners | Thousand US $ |
| *Y* | GDP in Iran and trade partners | Thousand US $ |
| *YP* | GDP per capita in Iran and trade partners | Thousand US $ |
| *DIS* | Distance between capitals of Iran and trade partners | Kilometers |
| *REMO* | Multilateral Resistance Term (MRT) | - |
| *CTI* | The CTI in Iran/the CTI in trade partner | % |
| *KAOPEN* | The KAOPEN in Iran/the KAOPEN in trade partner | % |
| *Sanctions* | Dummy variable taking a value of one if there are sanctions against Iran | Dummy (0/1) |

### 4.2. Model Specification

The earliest form of the gravity model which was introduced by Tinbergen (1962) [36] has the following structure:

$$lnExport_{ij} = \beta_0 + \beta_1 lnY_i + \beta_2 lnY_j + \beta_3 lnDIS_{ij} + \varepsilon_{ij},$$

---

[2] The study covers the period up to the voting by the Great Britain to withdraw from the EU.
[3] Anderson (2016) [30] expresses that "Gravity fits well with either aggregate or disaggregated trade flow data".
[4] Website link: http://web.pdx.edu/~ito/Chinn-Ito_website.htm.

where the export volume of country $i$ to $j$ ($lnExport_{ij}$) has a relationship with the GNP in country $i$ ($Y_i$) and in country $j$ ($Y_j$), meanwhile the distance between countries $i$ and $j$ ($DIS_{ij}$) as a proxy for transportation cost.

Over the years, numerous scholars have developed the above basic form by using other real or dummy variables. For instance, Linnemann (1966) [37] extends the gravity model and introduces population size of countries $i$ and $j$, and the artificial trade resistance factor. Frankel (1992) [38] uses the basic form further income (GDP per capita). Pfaffermayr (1994) [39] adds foreign direct investment as a variable affecting trade flows between countries. Cheng and Wall (2005) [40] uses the trade policy index and Nguyen (2010) [41] includes bilateral exchange rate and regional trade preference. Anderson and Wincoop (2003) [42] define the multilateral resistance factors (MRFs), such as language, remoteness, etc. Guttmann and Richards (2004) [43] include the openness level as a variable influencing trade between countries.

In this study, we further develop a gravity model, recently suggested by Yang et al. (2004) [24] to model bilateral trade flow between Iran and her trade partners in the EU and Asia to find out how sanctions can impact the bilateral trade of Iran with her trade partners.

As the first step to expanding the model of Yang et al. (2004) [24], we add the composite trade intensity (CTI) that was introduced by Squalli and Wilson (2006) [44] and is calculated as follows:

$$CTI_i = \frac{n \, (X + M)_i^2}{GDP_i \sum_{j=1}^n (X + M)_j},$$

where $X + M$ represents the trade of a country. The common trade openness formula contains a one-dimensional measures of trade openness, while the CTI considers both the relative position of a country's trade flow compared to her economic size and also the importance of a country's trade volume to world trade (Elmorsy (2015) [45]).

The second additional variable in our model is the Chinn–Ito index, which is an index measuring a country's financial openness. Since financial openness can be considered as an affecting factor on trade enhancement (Zhang and et al. (2015) [46] and Menyah and et al. (2014) [47], it would be a proper variable in our gravity model.

The last added variable to the model of Yang et al. (2004) [24] is the Multilateral Resistance Term (MRT). The MRT was firstly used by Anderson and Wincoop (2003) [42] in a gravity model as:

$$x_{ij} = \frac{y_i y_j}{y^w} \left( \frac{t_{ij}}{\pi_i P_j} \right)^{1-\delta},$$

where $x_{ij}$ represents nominal export from country $i$ to trade partner $j$, $y_i$ denotes nominal income of country $i$ and $y^w$ is world income ($=\sum_j y_j$). $\delta$ shows the elasticity of substitution across goods and $t_{ij}$ indicates international trade costs. Anderson and Wincoop (2003) [42] call $\pi_i$ and $P_j$ multilateral resistance variables. Baier and Bergstrand (2007) [48] express that the nonlinear estimation technique for the multilateral resistance factor in Anderson and Wincoop (2003) [42] is complex. According to Baier and Bergstrand (2007) [48], GDP weighted average of distance from trading partners can be used as a proxy for the multilateral resistance term.

Overall, the gravity model in our case can be written as follows, comprising only the time-variant variables as:

$$lnTRADE_{ijt} = \delta_1 + \delta_{2a}\ln\left(Y_{it}Y_{jt}\right) + \delta_{2b}\ln\left(YP_{it}YP_{jt}\right) + \delta_4 lnCTI_{ijt} + \delta_5 lnKAOPEN_{ijt} + \delta_6\ln REM_{ijt} + \varepsilon_{ijt},$$

where *TRADE* represents trade volume between Iran (country $i$) and a trading partner (country $j$) at specific time $t$. $Y_{it}Y_{jt}$ indicates the economy size of Iran and trading partner $j$ at time $t$. Moreover, $YP_{it}YP_{jt}$ shows income (GDP per capita) for Iran (country $i$) and a trading partner (country $j$), and *REM*,

*KAOPEN* and *CTI* show the MTR, financial openness and the composite trade intensity at time *t*, respectively.

According to Narayan and Nguyen (2016) [49] and Rasoulinezhad and Kang (2016) [50], to avoid the multicollinearity problem, it is better to break the above gravity model into two various models in which GDP and income variables are considered separately in each. Following this idea, the two following gravity models will be applied in our study:

Model I:

$$lnTRADE_{ijt} = \delta_1 + \delta_{2a}ln\left(Y_{it}Y_{jt}\right) + \delta_3lnCTI_{ijt} + \delta_4lnKAOPEN_{ijt} + \delta_5lnREM_{ijt} + \varepsilon_{ijt},$$

Model II:

$$lnTRADE_{ijt} = \delta_1 + \delta_{2b}ln\left(YP_{it}YP_{jt}\right) + \delta_3lnCTI_{ijt} + \delta_4lnKAOPEN_{ijt} + \delta_5lnREM_{ijt} + \varepsilon_{ijt}.$$

The above two gravity equations only comprise time-variant variables. Similarly to other gravity frameworks, our models have some time-invariant variables, i.e., distance (real variable) and sanctions (dummy variable):

$$\text{Time invariant variables: } \delta_5lDIS_{ij} + \delta_6sanctions.$$

Here, $DIS_{ij}$ indicates the distance between capitals in Iran (country *i*) and a trading partner (country *j*). Meanwhile, the variable "*sanctions*" is a dummy variable, which is captured bi-nominal variables. It takes a value of 1 if there are sanctions against Iran, or takes 0 otherwise.

Since we will have four different panel data estimations (Model I, II in the case of Iran–EU bilateral trade and Model I, II in the case of Iran–Asian countries' bilateral trade), the expected signs of coefficients in our gravity models can be explained as in Table 2.

**Table 2.** Expected signs of the variables.

| Variable | Type | Expected sign |
|---|---|---|
| *Trade* | Time-variant | Positive |
| $Y_{it}Y_{jt}$ | Time-variant | Positive |
| $YP_{it}YP_{jt}$ | Time-variant | Positive |
| *REM* | Time-variant | Positive |
| *CTI* | Time-variant | Positive |
| *KAOPEN* | Time-variant | Positive |
| *Dis* | Time-invariant | Negative |
| *Sanctions* | Time-invariant, Dummy | Positive (trade with Asia) |
| *Sanctions* | Time-invariant, Dummy | Negative (trade with Europe) |

Source: Authors' compilation.

According to the theoretical framework of the gravity model, it is expected that economy size and income would have positive impacts on trade volume and encourage trade between Iran and her trading partners, including 25 members of the EU and 25 Asian countries. It is also expected that the coefficient of the openness level (either trade openness or financial openness) may be positive. In the case of MTR (REM), trade volume may be enhanced for the higher multilateral resistance of the exporter *i*. In regards to the time-invariant variables, the coefficient of DIS is expected to bear a negative sign as distance shows the transportation cost between Iran and a trading partner. Due to Iran's trade policy of Asianization and de-Europeanization, it is expected that the sign of sanctions would be negative in Iranian trade with Europe, but have a positive sign in trade flows between Iran and her Asian trade partners.

## 5. Results and Discussion

### 5.1. Panel Cross-Section Dependence Test

Before applying panel unit root tests, cross-section dependence should be tested to find out whether the sample data are cross sectional dependent or independent. Otherwise, based on Breusch and Pagan (1980) [51] and Pesaran (2004) [52], the results of our estimations would be biased and inconsistent. According to the time and cross sections in our study, the Pesaran (2004) [52] residual cross-section dependence (CD) test is computed based on the pairwise correlation coefficients $\hat{\rho}_{ij}$ as below:

$$CD = \sqrt{\frac{2}{N(N-1)}} \sum_{i=1}^{N} \sum_{j=i+1}^{N} \sqrt{T_{ij}} \hat{\rho}_{ij}.$$

Based on the results of the CD Pesaran (2004) test, shown in Table 3, the null hypothesis (no cross-section dependence in residuals) can be strongly rejected at the 5% level. It implies that all series have strong evidence for cross-sectional dependence.

**Table 3.** Pesaran (2004)'s CD Test.

| Case | Variables | Pesaran's CD Test | Probability |
|---|---|---|---|
| Iran–EU Trade | LTRADE | 14.43 | 0.00 |
| | LYY | 34.09 | 0.00 |
| | LYPYP | 29.89 | 0.00 |
| | LCTI | 36.02 | 0.00 |
| | LKAOPEN | 27.11 | 0.00 |
| | LREM | 25.73 | 0.0 |
| Iran–Asian Countries Trade | LTRADE | 18.28 | 0.00 |
| | LYY | 43.30 | 0.00 |
| | LYPYP | 31.59 | 0.00 |
| | LCTI | 37.25 | 0.00 |
| | LKAOPEN | 30.42 | 0.00 |
| | LREM | 28.12 | 0.00 |

Source: Authors' compilation from Eviews 9.0 (IHS Global Inc., Irvine, CA, USA).

The results of the cross-section dependence test show which kind of panel unit root test is appropriate to apply. For cross-sectional independence in panels, using the LLC test and PP test is more convenient because they assume cross-sectional independence. Based on our finding which depicts cross-sectional dependence of our series, the most proper unit root test is the cross-sectionally augmented ADF [53].

### 5.2. Panel Unit Root Tests

In order to determine the stationarity of all the underlying time series data in a cross sectional dependent panel, we carry out the CADF panel unit root test (Pesaran (2007) [53]) for the variables at levels and first differences.

Pesaran (2007) [53] for a panel with $N$ cross-sectional units and $T$ time series observations, suggests a simple linear heterogeneous model as:

$$Y_{i,t} = (1 - \delta_i)\mu_i + \delta_i Y_{i,t-1} + u_{i,t}, \ i = 1, \ldots, N, \ t = 1, \ldots, T,$$

and suggests a test based on the t-ratio in the following cross-sectionally ADF regressions:

$$\Delta Y_{i,t} = a_i + b_i Y_{i,t-1} + c_i \overline{Y}_{t-1} + d_i \Delta \overline{Y}_t + \epsilon_{i,t}.$$

In the above equation, $\overline{Y}_t = \frac{1}{N}\sum\limits_{i=1}^{N} Y_{i,t}$ and $\Delta\overline{Y}_t = \frac{1}{N}\sum\limits_{i=1}^{N} \Delta Y_{i,t}$. Furthermore, $\epsilon_{i,t}$ indicates the regression error.

By applying this unit root test through the software, the results are calculated as reported in Table 4:

**Table 4.** Panel unit root test results.

| Case | Variable | Pesaran's CADF | H0 | Stationary |
|------|----------|----------------|-----|-----------|
| Iran–EU Trade | *LTrade* | 19.55 (0.81) | Accept | No |
| | *D (LTrade)* | 325.49 (0.00) | Reject | Yes |
| | *LYY* | 23.02 (0.63) | Accept | No |
| | *D (LYY)* | 200.83 (0.00) | Reject | Yes |
| | *LYPYP* | 2.94 (1.00) | Accept | No |
| | *D (LYPYP)* | 232.52 (0.00) | Reject | Yes |
| | *LCTI* | 25.74 (0.69) | Accept | No |
| | *D (LCTI)* | 264.85 (0.00) | Reject | Yes |
| | *LKAOPEN* | 20.02 (0.86) | Accept | No |
| | *D (LKAOPEN)* | 249.13 (0.00) | Reject | Yes |
| | *LREM* | 18.66 (0.52) | Accept | No |
| | *D (LREM)* | 198.11 (0.00) | Reject | Yes |
| Iran–Asian Countries Trade | *LTrade* | 24.12 (0.50) | Accept | No |
| | *D (LTrade)* | 193.28 (0.00) | Reject | Yes |
| | *LYY* | 9.83 (0.93) | Accept | No |
| | *D (LYY)* | 259.01 (0.00) | Reject | Yes |
| | *LYPYP* | 14.24 (0.53) | Accept | No |
| | *D (LYPYP)* | 301.62 (0.00) | Reject | Yes |
| | *LCTI* | 23.09 (0.59) | Accept | No |
| | *D (LCTI)* | 277.10 (0.00) | Reject | Yes |
| | *LKAOPEN* | 24.69 (0.62) | Accept | No |
| | *D (LKAOPEN)* | 213.84 (0.00) | Reject | Yes |
| | *LREM* | 20.02 (0.83) | Accept | No |
| | *D (LREM)* | 186.19 (0.00) | Reject | Yes |

Note: Numbers in brackets indicate *p*-values. Source: Authors' compilation from Eviews 9.0.

The reported *p*-values in the above table imply that all the series are non-stationary at levels (meaning accepting the null hypothesis representing the series containing a panel unit root) and stationary (rejecting the null hypothesis) at their first difference, which stands for the integration at I(1).

*5.3. Pedroni Panel Cointegration Test*

Since all the variables are cointegrated at I(1), the Pedroni panel cointegration test can be applied to find out whether there is any long-run equilibrium relationship between the series (Taghizadeh Hesary et al. (2015) [54] and Nasre Esfahani and Rasoulinezhad (2016) [55]). The achieved results are presented in Table 5. From the results of all the panel tests, most statistics have *p*-value less than 0.05, and, hence, the majority of the all statistics tests can significantly reject the H0 of no cointegration at the 5% significance level. In summary, it can be concluded that there is evidence of a long-run relationship between variables in all our four models.

**Table 5.** Pedroni Panel Cointegration Test results.

| Case | Model | | Statistic | Probability | Weighted Statistic | Probability |
|------|-------|---|-----------|-------------|--------------------|-------------|
| Iran–EU Trade | Model I | Panel v-statistic | −0.13 | 0.55 | −3.34 | 0.99 |
| | | Panel rho-statistic | 3.97 | 1.00 | −2.45 * | 0.00 |
| | | Panel PP-statistic | −0.49 | 0.30 | −3.97 * | 0.00 |
| | | Panel ADF-statistic | −4.49 * | 0.00 | −4.27 * | 0.00 |
| | | Group rho-statistic | 5.46 | 1.00 | - | - |
| | | Group PP-statistic | −3.81 * | 0.00 | - | - |
| | | Group ADF-statistic | −4.47 * | 0.00 | - | - |
| | Model II | Panel v-statistic | 1.53 ** | 0.06 | 0.79 | 0.21 |
| | | Panel rho-statistic | −4.36 * | 0.00 | −2.81 * | 0.00 |
| | | Panel PP-statistic | −5.81 * | 0.00 | −4.74 * | 0.00 |
| | | Panel ADF-statistic | −5.13 * | 0.00 | −4.55 * | 0.00 |
| | | Group rho-statistic | −2.20 * | 0.01 | - | - |
| | | Group PP-statistic | −5.56 * | 0.00 | - | - |
| | | Group ADF-statistic | −5.31 * | 0.00 | - | - |
| Iran–Asian Countries Trade | Model I | Panel v-statistic | 2.05 * | 0.02 | −0.78 | 0.78 |
| | | Panel rho-statistic | −1.73 * | 0.04 | −0.34 | 0.36 |
| | | Panel PP-statistic | −6.76 | 0.00 | −4.20 * | 0.00 |
| | | Panel ADF-statistic | −7.38 * | 0.00 | −4.81 * | 0.00 |
| | | Group rho-statistic | 0.69 | 0.75 | - | - |
| | | Group PP-statistic | −6.23 * | 0.00 | - | - |
| | | Group ADF-statistic | −5.98 * | 0.00 | - | - |
| | Model II | Panel v-statistic | 1.05 | 0.14 | 0.51 | 0.30 |
| | | Panel rho-statistic | −2.89 * | 0.00 | −2.02 | 0.02 |
| | | Panel PP-statistic | −4.36 * | 0.00 | −3.97 * | 0.00 |
| | | Panel ADF-statistic | −3.94 * | 0.00 | −4.27 * | 0.00 |
| | | Group rho-statistic | −1.64 * | 0.05 | - | - |
| | | Group PP-statistic | −4.49 * | 0.00 | - | - |
| | | Group ADF-statistic | −4.91 * | 0.00 | - | - |

Note: (*) and (**) show statistical significance at the 5% and 1% level, respectively. Source: Authors' compilation from Eviews 9.0.

*5.4. Gravity Model Estimation*

After applying the cointegration test and finding out that there is a long-run relationship between series in all our gravity equations, the three panel data estimation approaches, i.e., fixed effect (FE), random effect (RF) and fully modified OLS (FMOLS) is applied to explore the coefficients of our all variables. Due to the fact that there is not a similar view to the estimation of panel co-integration (for instance, Pedroni (1996 [56], 2001 [57]) recommend the fully modified OLS (FMOLS) estimator. Cheng and Wall (2005) [40] and Anderson and Wincoop (2003) [42] suggest the fixed effects (FE) or Soren et al. (2014) [58] propose the random effects (RE) because FE does not allow for the time-invariant real variables in a gravity model. Fidrmuc (2009) [59] believes that since many macroeconomic variables like GDP are most likely I(1), there is not any problem with using fixed or random effects estimators, and their results are similar to the fully modified OLS.). Therefore, we apply all of these three panel estimators to find and compare results. It should be mentioned that the coefficients for the time-invariant real variables, i.e., distance, can not be estimated by the FE estimator. The findings are reported in Table 6.

As it can be seen, the basic features of gravity model estimations are very similar across all three estimators. The estimation results of "Model I" for the bilateral trade of Iran-25 EU member countries confirm that GDP, trade openness (the composite trade index), financial openness (The Chinn–Ito index) and MTR have a significant positive impact on Iran–EU bilateral trade, while distance negatively influences the trade volume. Moreover, as we predicted, sanctions against Iran decrease the trade volume of this country and the EU member states. This result proves the Iran's trade policy of de-Europeanization.

**Table 6.** The gravity model estimation.

| Case | Model | Variables | FE | RF | FMOLS |
|------|-------|-----------|-----|-----|-------|
| Iran–EU Trade | Model I | LYY | 0.26 (0.08) | 0.17 (0.01) | 0.38 (0.00) |
| | | LCTI | 0.43 (0.00) | 0.38 (0.00) | 0.48 (0.01) |
| | | LKAOPEN | 0.32 (0.04) | 0.31 (0.00) | 0.41 (0.00) |
| | | LREM | 0.25 (0.00) | 0.23 (0.02) | 0.31 (0.01) |
| | | LDIS | - | −1.08 (0.05) | - |
| | | SANC | −0.56 (0.00) | −0.48 (0.00) | −0.57 (0.00) |
| | Model II | LYPYP | 0.29 (0.03) | 0.14 (0.05) | 0.50 (0.09) |
| | | LCTI | 0.39 (0.02) | 0.40 (0.00) | 0.43 (0.00) |
| | | LKAOPEN | 0.36 (0.00) | 0.32 (0.03) | 0.39 (0.00) |
| | | LREM | 0.09 (0.01) | 0.08 (0.00) | 0.12 (0.01) |
| | | LDIS | - | −1.37 (0.00) | - |
| | | SANC | −0.75 (0.00) | −0.69 (0.00) | −0.76 (0.00) |
| Iran–Asian Countries Trade | Model I | LDYP | 0.50 (0.00) | 0.50 (0.00) | 0.49 (0.00) |
| | | LCTI | 0.53 (0.01) | 0.49 (0.04) | 0.58 (0.00) |
| | | LKAOPEN | 0.42 (0.00) | 0.40 (0.00) | 0.48 (0.01) |
| | | LREM | 0.17 (0.00) | 0.15 (0.00) | 0.23 (0.00) |
| | | LDIS | - | −2.90 (0.00) | - |
| | | SANC | 0.31 (0.00) | 0.29 (0.00) | 0.46 (0.00) |
| | Model II | LYPYP | 0.14 (0.02) | 0.12 (0.04) | 0.19 (0.00) |
| | | LCTI | 0.47 (0.03) | 0.46 (0.02) | 0.51 (0.02) |
| | | LKAOPEN | 0.45 (0.00) | 0.44 (0.00) | 0.48 (0.00) |
| | | LREM | 0.11 (0.04) | 0.09 (0.04) | 0.15 (0.00) |
| | | LDIS | - | −1.78 (0.03) | - |
| | | SANC | 0.84 (0.00) | 0.83 (0.00) | 0.96 (0.00) |

Source: Authors' compilation from Eviews 9.0.

The estimation findings of "Model II" for the trade of Iran-25 EU member states depict that income (GDP per capita), trade openness (the composite trade index), financial openness (the Chinn–Ito index) and MTR increase the bilateral trade volume between Iran and the 25 EU member states, while, similar to the first model estimation result, distance and sanctions have a significant negative impact on the trade volume.

In the case of Iran's bilateral trade with the 25 Asian countries, the results reveal that a 1% increase in the joint GDP of Iran and the 25 Asian countries, raises the bilateral trade volume by approximately 0.50%. Joint income (GDP per capita) has a less positive influence on the Iran–Asian countries' bilateral trade. The results show that the bilateral trade between these countries is boosted up about 0.15% with a 1% increase in the joint GDP per capita. Moreover, the effect of the sanctions (SANC) on trade is positive and significant, which confirms the ongoing Asianization of Iran's trade policy. In addition, trade openness, financial openness and MTR have positive effects on Iran–Asian trade.

The findings of the models' estimations provide evidence of a significant negative effect of sanctions on Iran–EU member states' bilateral trade. The coefficient of SANC is estimated at an average of 48% (=Exp(−0.65) − 1) by FE estimator, compared to an average of 44.2% (=Exp(−0.58) − 1) by FE and 48.5% (=Exp(−0.66) − 1) by FMOLS. This indicates that trade volume decreases by nearly 46.9%[5] when the sanctions are imposed against Iran.

In regards to the positive effect of sanctions on Iran–Asian countries' bilateral trade, it can be calculated that the trade volume increases by about 77.7% (=Exp(1.15) − 1) by FE, 75% (=Exp(1.12) − 1) by RE and 103% (=Exp(1.42) − 1) by FMOLS. As an average of findings by these three estimators, trade volume between Iran and the 25 Asian countries would increase by 85.2%[6].

---

[5]　It is calculated as the average of 44%, 44.2% and 48.5%.
[6]　It is calculated as the average of 77.7%, 75% and 103%.

In the case of distance as a proxy of transportation cost, the negative sign of its coefficient, estimated by random effect (RE), represents that geographical distance has a negative impact on bilateral trade between Iran and the EU member states and Asian countries. A 1% increase in this variable decreases the trade volume between Iran and the 25 EU member states and 25 Asian countries by an average of 1.22%[7] and 2.34%[8], respectively.

## 6. Conclusions

This study mainly tried to empirically find whether the imposition of sanctions against Iran has pushed this country toward Asianization and de-Europeanization. To this purpose, we investigated the impact of GDP, GDP per capita, trade openness, financial openness, MTR, distance and sanctions on Iran–EU members' and Asian countries' bilateral trade through the estimations of a gravity model from 2006 to 2013. Following Narayan and Nguyen (2016) [49], we developed different gravity model equations according to GDP and GDP per capita to avoid any multicollinearity problem. The estimations of these equations were done by three panel approaches, i.e., fixed effect, random effects and the fully modified OLS.

Our estimation results were in line with the opinion of Fidrmuc (2009) [59] about similarity of estimators' results for panel co-integration. The results revealed that the basic features of gravity model estimations are very similar across all three estimators, i.e., FE, RE and FMOLS.

The empirical results showed that an increase in GDP implies an increased trade flow between Iran and the trade partners in both the EU and Asia. Furthermore, the positive effect of income on the Iran–EU member states' bilateral trade (by an average of 0.31%) is higher than the positive effect of income on the Iran–Asian countries' bilateral trade (by an average of 0.15%). This result proves Staffan Linder's theory (Linder (1961) [60]). He expressed that individuals with different income levels tend to consume various bundles of goods with richer consumers expressing a latent demand for more goods. In our study case, since most countries in the EU are developed, consumers are rich enough to be able to afford product varieties. This fact is reflected by the higher coefficient of income for the Iran–EU members' bilateral trade rather than the Iran–Asian countries' bilateral trade.

In addition, the results revealed that the trade–distance nexus is negative for both Iran–EU members and Iran–Asian countries. This finding is in line with many previous studies such as Leamer (2007) [61] and Disdier and Head (2008) [62], who found that trade volume declines dramatically with the distance. This variable can be considered as a geographical barrier between two trading partners and also as a cost for transportation.

In the case of trade openness, the positive impact of this variable on the Iran–EU members' bilateral trade (by an average of 0.41%) is less than the positive effect of it on the trade flows between Iran and Asian countries (by an average of 0.50%). The situation for financial openness is similar as well. The average coefficient of this variable in the case of Iran–EU trade (0.35%) is less than its average for Iran–Asia bilateral trade (0.44%). However, MTR, which was used by a proxy of GDP weighted average of distance from trading partners, conversely, differs. This means that its average coefficient for the bilateral trade between Iran and Asian countries is less than for Iran–EU trade.

In regards to sanctions, the empirical estimations proved the negative effect of this variable on the Iran–EU bilateral trade (by an average of 46.9%), while it has a positive impact on trade between Iran and Asian countries (by an average of 85.2%). In total, these findings, which are in line with Borszik (2015) [28] and Haidar (2016) [29], empirically confirmed that the imposition of various sanctions related to Iran's nuclear program has pushed the foreign trade policy of this country towards Asianization and away from Europeanization.

---

[7] It is calculated as the average of 1.08 and 1.37.
[8] It is calculated as the average of 2.90 and 1.78.

It can be summed up that sanctions pushed Iran to modify her foreign trade policy in the direction of increasing trade ties and relations with Asian countries who have become new trade partners for Iran instead of the European Union. Hence, sanctions can be considered as an instrument for modification of foreign trade of a target country against whom a sender or a group of senders imposes sanctions or trade restrictions. However, upon modifying foreign trade direction, a target country may face a higher transportation cost (as proved to be the case for Iran). As a trade policy, a target country has to pay attention to trade costs, which can lower the importer/exporter's profit and the bilateral trade flow as well. Moreover, a target country may find a new trading partner with high-income levels, which shows the magnitude of local demand. According to the Linder theory, people in a high-income country have a tendency to consume more varieties of importing goods rather than people in a low-income country. It was proved in our research by the higher coefficient of income for the Iran–EU trade, rather than the Iran–Asian countries' trade volume.

Taken together, it can be noted that obviously there are many other factors such as geopolitical concerns, Iran's situation towards joining the WTO, tariffs and pricing, and visa procedures and transports, which have significant impact on the Iran–EU or Iran–Asian countries' trade. The authors suggest future research with more data related to these factors, giving a better result and fewer errors. Furthermore, future avenues of research should consider the estimation of the gravity model with some other variables for Iran's export and import separately. However, from our point of view, this research proves useful and interesting findings, which can help economists and policy makers to achieve a better view of Iran's bilateral trade with EU members and Asian countries as well.

**Acknowledgments:** The authors would like to thank the anonymous reviewers for their helpful and constructive comments that greatly contributed to improving the final version of the paper. They would also like to thank the Editor for the generous comments and support during the review process.

**Author Contributions:** Both authors contributed equally to this work.

**Conflicts of Interest:** The authors declare no conflict of interest.

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
