# Peer review of "Have Sanctions Modified Iran’s Trade Policy? An Evidence of Asianization and De-Europeanization through the Gravity Model"

_economies, doi:10.3390/economies4040024_

Reviewer 1 Report

According to the introduction, the paper should aim at assessing whether the EU sanctions on Iran have diverted Iranian trade flows from Europe to Asian countries. The results of this exercise can provide an additional hint on whether economic sanctions are effective tools to persuade a country to modify its policy. The research question is definitely interesting and the research on the Iranian case is limited but not inexistent. The interest on this topic is also relevant and applicable to other cases, like the sanctions imposed by EU and US on Russia, even if the Iranian case allows a better estimation of it effects due to longer time horizon. Notwithstanding the interesting research question, the paper shows some weaknesses mainly in the structure and in the literature review. Those parts can be definitely improved.

Indeed, it is not clear whether the author wants to analyze the effectiveness of the sanctions imposed by the EU in Iran or, more generally assessing the evolution of the Iranian trade policy during the sanctions period. In the first case, then I would have paid much more attention in the literature review to previous studies on the role of the sanctions in trade flows and not only in Iran. Beside the main contributions like Hufbauer, Schott, Elliott (2009) and Evenett  (2002), more specifically, I would have mentioned the whole literature assessing the impact of trade shock on export dynamics (Iacovone and Javorcik, 2010 , for instance). If, instead, the aim is simply running an extended gravity equation to analyze trade flows to judge the trade diversion, a better overview of all the  explanatory variables that can affect them (other than sanctions) is needed.

There are then more specific comments on the structure of the paper. First, the introduction could incorporate the second section in order to provide a clearer background analysis without mentioning the tested hypothesis ( this normally comes much later in the paper). I would then focus the literature review on the methodologies applied by previous studies on the effect of sanctions also from a geopolitical point of view, as suggested before. The alternative hypothesis of the second test is wrong (line 59): indeed, it should be ‘there is not a positive relationship…’. On data, Table 2, it is not specified to which degree data on trade flows have been disaggregated. Moreover, I would have extended the gravity equation by adding other explanatory variables suggested by the literature like oil price for instance. On methodology, there are few studies that adopt different methodologies to understand the impact of sanctions: in particular, Haidar (2014) assessed the effect of the sanctions in Iran on trade diversion through exporters  firm- level analysis by shedding  light on the exporter dynamics. I would at least incorporate these results to confirm what the gravity model suggests.

Author Response

First of all, we would like to thank you for your comments and suggestions which we found very helpful for improving our manuscript. We made our manuscript succinct and carefully revised it based on your valuable comments. Besides, we have mentioned some of your valuable commends as suggestions for future researches in the “Conclusion” part.

Our specific changes are listed below:

1-     We modified and re-wrote some parts of our paper, i.e. Introduction, literature review and empirical research

2-     We corrected our error in writing hypotheses in the introduction paper.

3-     We corrected our Table 1.

4-  We considered some new related references such as Haidar (2016) in the empirical researchers.

All the rest of our changes, particularly those to improve the English language of our manuscript are shown in the attached revised manuscript in the red color.

Thanks again.

Reviewer 2 Report

Manuscript: Have sanctions changed Iran’s trade policy? An evidence of Asianization and de –Europeanization through the gravity model.

 Reviewer comments

The paper seeks to examine the impact of sanctions on Iranian trade and also to assess whether the imposition of Western sanctions on Iran has shifted the focus of trade policy towards Asia.

The topic is relevant topic considering that the cases of sanctions have been increasing in recent times. 

1.The author must thoroughly review or edit the language. This makes it difficult to follow the article. It will be best if they can get a professional to edit the article.

2. There is no need for stating both the null and alternative hypotheses in current articles. It suffices to state only the null.

3. The use of endnote references makes it diffiuct to start a sentence in the article with a reference. For example; how you use references 24, 25, 26 etc. in-text. Page 5

4. In the gravity model, summing up export and import is not current standard practice. It will be better to estimate the gravity model for either of the two separately.

5. For the MTR, there is a more theoretically accepted appraoch to computing it than using  GDP weighted distance. See Bair and Bergstrand (2009).

6. On page 8, you indicated that sanction dummy variable was time-invariant, so I was expecting that this will also be differenced away  as distance in the FE estimations.

7. In the estimations, the GDP elasticity of trade were not close to unity, but we know that is expected to approximately close to one.

8. There is also no theoretical framework to underpin the empirical work. How is sanction theoretically expected to affect trade flow between the sender and the target?

References:

Baier, S. L., & Bergstrand, J. H. (2009). Bonus vetus OLS: A simple method for approximating international trade-cost effects using the gravity equation.Journal of International Economics77(1), 77-85.

Author Response

We would like to thank you for your comments and helpful suggestions. We
revised our manuscript according to these comments and suggestions. Below are the answers to each specific points.

1-     We edited and proofread the manuscript to improve its English language level.

2-     We revised state of hypotheses and only wrote the null hypotheses.

3-     We corrected the references and citations in the text.

4-     Since, we have followed the gravity model of Narayan and Nguyen (2016) in which includes the aggregate trade volume, our model does not consider export and import separately. But we have suggested this valuable tip for the future researches in the “Conclusion” part.

5-     We checked Bair and Bergstrand (2009) for their theoretically-motivated exogenous multilateral resistance terms. Since their new gravity model is based on a Taylor-Series expansion, we have recommended its structure and MRT for the future research and mentioned it in the “Conclusion” part.

6-     In the case of “sanctions”, since it is a dummy variable , we can have estimation of it through FE. But the distance which is a real time-invariant variable, its estimation through FE is not impossible. For clarification, we added the word “Dummy” for sanctions in Table 2.

7-     In a normal economic situation, the elasticity of trade to GDP is close to unity. But in our case of Iran as an natural resource based economy under several sanctions, the low elasticity of trade to GDP is common. In fact, the most factors explaining the lower elasticity of trade in Iran are changes in the composition of demand under sanctions, weak trade finance and increased trade protection.

8-     We re-wrote the literature review,  which theoretically explains how sanctions affect the trade flow between the sender and the target.

All the corrections and changes are shown in red color in the attached revised manuscript.

Sincerely Yours.

Round  2

Reviewer 2 Report

The paper still needs a bit of editing and polishing up.

eg, page 5 "The second group of the literature is resulted from the attempts of researchers to find out the  effects of sanctions by using a gravity model" Suggestion. The second strand of literature attempted to find out the effect of sanctions using the gravity model.

page " Raul Caruso (2005) [31] supposes economic sanctions as quantitative restrictions" This should rather be: Caruso (2005) indicated that economic sanctions (may) act as quantitative restrictions

There are many more of the above errors.

The history of sanctions in the theoretical section is not important (I suggest you remove the first three paragraphs of that section)

It will be better to use one in-text referencing style. I have seen you use the name, number or both. I think you should be consistent with names plus the numbers. 

Also, in the references, you must bring the surname first (before the initials) and this should be arranged alphabetically. 

In addition why do you use Raul Caruso and also Caruso for your in-text references. Be consistent and use only Caruso.

Author Response

We thank the anonymous reviewer for his/her careful reading of our manuscript and giving us valuable comments and suggestions. Herein, we explain how we have revised again the paper based on precious reviewer's comments.

1-  In page 5, we replaced the sentence to "The second strand of literature attempted to find out the effect of sanctions using the gravity model"

2- In our manuscript at page.2 , we changed the sentence to "Caruso (2005) indicated that economic sanctions may act as quantitative restrictions"

3- We removed the first three paragraphs about the history of sanctions in the Theoretical Framework part at page.2 

4- We reconstructed the referencing style in the text based on the names plus the numbers.

5- In Reference part,  we brought  the surname first.

6- We corrected using Caruso in our text.

7- We did the proofreading to improve the English language level of our text.

It should be mentioned that all our new changes in the text have been shown in red color.

Thanks again.